# Temporal Associations Between Sport Participation, Dropout from Sports, and Mental Health Indicators: A Two-Year Follow-Up Study

**DOI:** 10.3390/bs15121665

**Published:** 2025-12-02

**Authors:** Reidar Säfvenbom, Tommy Haugen, Vidar Sandsaunet Ulset, Andreas Ivarsson

**Affiliations:** 1Department of Teacher Education and Outdoor Studies, Norwegian School of Sport Sciences, 0863 Oslo, Norway; vidarsu@nih.no; 2Department of Sport Science and Physical Education, University of Agder, 4604 Kristiansand, Norway; tommy.haugen@uia.no; 3School of Health and Welfare, Halmstad University, 301 18 Halmstad, Sweden; andreas.ivarsson@hh.se

**Keywords:** youth, mental health, well-being, sport, informal movement activities

## Abstract

Participation in organized sports may have a positive effect on mental health, but documentation is based on methodological limitations. The objective of this study was to investigate the influence of dropout from organized youth sport on change in mental health and subjective well-being. A total of 1046 young persons (13–18 years old), all participating in organized sport at baseline, were included. Data on mental health and subjective well-being were collected at baseline and again after two years. Participants who were into organized sports and participants who reported other types of informal movement activity two years later reported less increase in negative affect over time compared to participants who had dropped out of sports and did not get involved in other types of movement activity. For positive affect, there was a credible difference in change between those who dropped out of organized sport but did not get involved in an alternative physical activity and those who were still active in organized sports. The results indicate that dropout from traditional youth sports might have a negative influence on well-being, but involvement in other types of informal and negotiable movement contexts might buffer the potential negative effects.

## 1. Introduction

Good mental health is fundamental for human functioning, enabling individuals to cope with the challenges of everyday life, realize and develop abilities, seize opportunities, learn, engage meaningfully in social contexts, and contribute to their communities ([10]). While mental health contributes to cohesion, economic productivity, and societal resilience ([23]), mental health problems may undermine educational attainment, limit participation in the labor market, and contribute to social exclusion ([1]; [5]; [34]). In the adolescent population, mental health plays a particularly important role, and it is therefore worrying that recent research highlights a particularly marked decline in mental well-being among young people ([4]; [15]; [35]). Global estimates indicate that mental disorders affect approximately 10% of children and adolescents worldwide.

Mental health in youth is influenced by a relatively complex interaction between individuals and their environment, where home, school, and leisure activities all play a role. It is crucial that young people also have access to protective factors ([17]) and developmental assets ([19]) outside their homes and schools. It should be emphasized that mental health comprises both the absence of psychological difficulties and the presence of positive psychological resources. Recent work shows that mental ill-being and subjective well-being are closely related yet partly independent dimensions, making it essential to examine both factors to accurately capture adolescents’ mental health ([29]).

Traditional youth sports have been considered as developmental assets, and participation in organized youth sports is related to increased levels of good mental health, life satisfaction, and emotional well-being, as well as reduced levels of mental illness symptoms ([8]; [26]). Despite the negative consequences ([30]), such as increased stress levels related to pressure to perform ([21]; [27]) and increased levels of burnout due to early specialization ([3]), organized sport contexts are seen as safe harbors that reduce distress symptoms and increase subjective well-being ([26]). In line with such relatively vague associations, governments and policymakers have funded national sport federations and promoted traditional sport contexts to support developmental processes in individuals and to combat health problems ([18]).

There is no single reason why traditional and organized youth sports should affect mental health or well-being. Youth sport contexts may, in various ways, provide biological, mental, and social support, which may lead to perceived competence, belonging, identity, autonomy, etc. in youth ([12]), but the relational complexity and the temporality issue have been questioned. However, a recent prospective study from [11] ([11]) gives valuable insight into the variance in sport contexts as well as the directionality of the relationships between sport participation and adolescents’ mental health. Their longitudinal study on a large number of Australian adolescents showed that team sport participation prospectively predicts fewer symptoms of depression and anxiety, yet not vice versa. No similar effects were detected for individual sport. The study also showed that more team sport involvement prospectively predicts fewer emotional symptoms, while more emotional symptoms predict less team sport involvement, and that more emotional symptoms prospectively predict less individual sport involvement, “but not vice versa” (p. 10). Finally, models for all the mental health indices in the studies were shown to be invariant by sex, except for depression. The association between sport participation and depression differed significantly for boys and girls (for both team and individual sports). Overall, the study revealed relatively complex interactions between different indicators of mental health and well-being and individual sports vs. team sports, and females vs. males.

With the exception of [11] ([11]), just a few studies (e.g., [32]) have adopted a repeated measure design to investigate the potential effects of change in sport participation ([7]). This is problematic because when cross-sectional studies show positive associations between mental health and participation in sports, these studies suggest that engaging in sports is linked to positive effects on mental health and that dropping out will have negative consequences. Currently, there is insufficient documentation to robustly support these findings. In addition, to the best of the authors’ knowledge, no longitudinal studies have compared the potential change in mental health and well-being between those who drop out of organized sport but still engage in other types of movement contexts and those who completely terminate their participation in any type of movement activity. Given that participation in informal, negotiable ([24]), and often self-organized movement activities have also been shown to have a positive influence on health and well-being ([28]), it is of interest to compare changes in mental health between this group and the other two groups (those who dropped out from organized sport and did not get involved in any movement activity and those who remained active in traditional and organized sport).

To address limitations in previous studies and extend the knowledge about the potential effects of youth sport attrition on mental health and subjective well-being, the objective of this study was to investigate the influence of dropout from organized youth sport on change in mental health (i.e., distress symptoms) and subjective well-being (i.e., positive affect and negative affect) outcomes over a two-year period. A second objective was to investigate the potential buffer effects that participation in informal, negotiable, and more self-organized movement contexts might have for adolescents who leave organized sports.

## 2. Materials and Methods

### 2.1. Participants and Procedures

The participants in the present study are from a larger research project entitled “the RElevance of Physical Activity Contexts in the everyday life of adolescents” (REPAC) ([9]). This study comprised two birth cohorts of adolescents from 44 different schools in Norway completing a survey three times in a two-year period. For this study, we decided to use data at the end of the first (T1) and the third (T2) school years at lower and upper secondary schools, respectively. The data collection was approved by the Norwegian Centre for Research Data (project number 37624). Written and oral information about the study was provided, and consent to participate was obtained from all participants. For participants younger than the age of 15, parental consent was also obtained. From the initial pool of participants at T1 (*N* = 2854), a total of 1562 participants (girls = 721, boys = 841), all participating in organized sport, were included (the other participants did not participate in organized sport when the first wave of data was collected). Of the 1562 organized sport participants who responded to the questionnaire at T1, 67% (*n* = 1046) also responded at T2. They were, therefore, included in the analyses. A sensitivity analysis to compare participants who dropped out of the study between the first and second wave of data collection and the participants who completed the data collection process at T2 was performed. The results showed trivial to small effects between the groups for the three health and well-being variables (Cohen’s d ranged between 0.06 and 0.20). At T1, the participants reported approximately 5–7 h of organized sports per week. Some of the most common sports were football, handball, floorball, track and field, swimming, and cross-country skiing. Before the analyses, the participants were assigned to one of three groups: (a) those who, during the last two years, had dropped out of organized sport and had not gotten involved in other types of physical activities (dropouts: *n* = 120); (b) those who had dropped out of traditional organized sport but were active in other types of informal movement activities (converters: *n* = 285); and (c) those who were still active within organized sport (stayers: *n* = 641).

### 2.2. Instruments

Hopkins Symptom Checklist (HSCL): The HSCL ([6]) was used to measure psychological distress symptoms (i.e., symptoms of anxiety and depression). We used the short Norwegian version consisting of 10 items, which has shown satisfying psychometrics ([14]). The participants were asked to indicate how much the listed symptoms had bothered and distressed them during the last week. All items were answered on a 4-point Likert scale ranging from “not at all” to “extremely”. The scale showed adequate internal reliability (T1 α = 0.87, T2 α = 0.92).

Basic Emotions State Scale: The Basic Emotions State Scale ([33]) was used to measure the respondents’ overall positive and negative emotions, with six items for positive emotions (satisfaction, pleasure, happiness, interest, engagement, and immersion) and three items for negative emotions (fear, anger, and sadness). Responses were given on a Likert scale ranging from 1 (never) to 7 (always). Both subscales showed acceptable internal reliability (positive emotions T1 α = 0.83, T2 α = 0.88; negative emotions T1 α = 0.77, T2 α = 0.83).

### 2.3. Data Analysis

We performed Bayesian multigroup latent change score (LCS) ([20]) analyses to examine potential group differences between the participants belonging to the three different groups with regard to within-person changes in health and well-being. Bayesian statistics are used to incorporate prior knowledge or beliefs into the interpretation of data, allowing for updates to these beliefs as new evidence becomes available. In the LCS model, a latent change score represented the absolute change between the construct measured at T1 and T2. The potential scale reduction factor (PSRF) was used to assess the model convergence (PSRF ≈ 1 was considered as evidence of convergence) ([13]). Bayesian models were implemented using Markov chain Monte Carlo (MCMC) simulation procedures with a Gibbs sampler and specified a fixed number of 150,000 iterations (the first half was used as the burn-in phase, which is the default in Mplus). Model convergence was assessed using both statistical criteria (i.e., PSRF < 1.1) and visual inspection of trace plots to ensure that multiple chains converged toward a similar target distribution ([20]). We used the posterior predictive p (PPp) value and the 95% confidence interval (CI) to assess model fit. A well-fitting model should have a PPp value around 0.50 in combination with a symmetric 95% CI centering on zero. For each parameter, a credibility interval was estimated. If the 95% CI did not include zero, the null hypothesis was rejected as improbable, and the parameter estimate was considered credible ([36]). A difference score was estimated for each pairwise comparison to determine if there was a credible difference in change between two of the groups. For all parameters, default priors in Mplus were used ([22]).

## 3. Results

### 3.1. Descriptive Results

The results showed credible relationships, ranging from small to large, between almost all health and well-being variables. Only between negative and positive affects did two of the relationships not reach the suggested cut-off for small effects. The descriptive results for the three subgroups (“dropouts”, “convertors”, and “stayers”) are presented in Table 1.

### 3.2. Results from the Latent Change Score Analyses

#### 3.2.1. Psychological Distress Symptoms

The model showed adequate fit to data (PPp = 0.47, 95% CI = [−13.73, 17.27]). There were credible increases in psychological distress symptoms in all three groups: the dropout group (Δ = 0.17, 95% credibility interval = [0.04, 0.28]), the self-organized physical activity group (Δ = 0.22, 95% credibility interval = [0.14, 0.29]), and the organized sport group (Δ = 0.14, 95% credibility interval = [0.10, 0.18]). There were credible variances in change within all groups. The difference test showed no credible difference in change between any of the three groups. Please see Figure A1 in Appendix A.

#### 3.2.2. Positive Emotions

The model showed a good fit to data (PPp = 0.48, 95% CI = [−15.76, 17.03]). There was a negative credible change in positive affect in the dropout group (Δ = −0.31, 95% credibility interval = [−0.55, −0.07]) but not in the self-organized physical activity group (Δ = −0.09, 95% credibility interval = [−0.25, 0.07]) or in the organized sport group (Δ = −0.05, 95% credibility interval = [−0.15, 0.05]). There were credible variances in change within all groups. The difference test showed credible differences in change between the dropout group and the organized sports group (Δ = 0.25, 95% credibility interval = [−0.52, 0.00]), indicating less favorable change for the group that dropped out of organized sports and did not attend any other activity contexts. Please see Figure A2 in Appendix A.

#### 3.2.3. Negative Emotions

The model showed a good fit to data (PPp = 0.48, 95% CI = [−15.37, 15.67]). There was a positive credible change in negative affect in the dropout group (Δ = 0.43, 95% credibility interval = [0.19, 0.68]) but not in the self-organized physical activity group (Δ = 0.08, 95% credibility interval = [−0.08, 0.25]) or in the organized sport group (Δ = 0.09, 95% credibility interval = [−0.02, 0.20]). There were credible variances in change within all groups. The difference test showed credible differences in change between the dropout group and both the self-organized activity group (Δ = 0.34, 95% credibility interval = [0.06, 0.64]) and the organized sports group (Δ = 0.34, 95% credibility interval = [0.07, 0.61]). Please see Figure A3 in Appendix A.

## 4. Discussion

This study proves first of all that studying change in mental health during adolescence requires study designs that follow young people over time. The longitudinal analyses showed a general increase in levels of distress and negative emotions and a corresponding decrease in levels of positive emotions during adolescence. The overall negative changes registered over the period of two years were small but systematic and significant, and they all confirm prior research showing a negative trajectory in mental health and well-being during adolescence ([25]). The results support prior calls for actions that can reduce or eliminate the negative development. No differences in change in mental distress over the two years were seen between (1) those who remained active in traditional youth sports, (2) those who left this type of activity context but still spent time in informal and often negotiable types of movement activities, and (3) those who left sports and did not report any involvement in other movement activities or contexts. These results indicate that mental distress symptoms during adolescence do not change according to traditional and organized sport participation and are in line with the literature that considers symptoms of anxiety and depression as substantial and resistant to change compared to well-being variables such as emotions ([16]) and partly in line with the study from [11] ([11]), which showed that team sport participation predicts fewer symptoms of depression and anxiety but that participation in individual sports does not.

Our analyses revealed differences in the change in well-being variables between the groups. Analyses on changes in positive emotions (holistic emotions, such as satisfaction, pleasure, and happiness, and eudaimonic emotions, such as interest, engagement, and immersion) showed that the group of adolescents who remained active in organized sports had a more favorable development in comparison to the group who totally terminated their sport participation and reported no involvement in other movement contexts. No difference in change was seen between the group who remained active in organized sports and the group who dropped out of sports but remained active in self-organized movement contexts. For negative emotions (fear, anger, and sadness), the results showed that the group who remained active in organized sport contexts, as well as the group who participated in self-organized movement contexts, had a more favorable development in comparison to the group who terminated sport participation and reported no involvement in movement activity.

There are three possible ways to explain the differences in well-being change identified in the present study: It may be caused by (1) the dropout itself, meaning that those who drop out from organized sports and are not involved in other types of informal movement activity do experience more negative emotions and less positive emotions after dropping out of sports; (2) other stressors that appeared in the adolescents’ everyday life before the dropout and could have caused both a decline in mental health and the dropout from sports; or (3) negative experiences of sport participation before the dropout, meaning that their well-being would have been even more reduced if they had kept on with their sport. These possibilities, and the latter one in particular, must be taken into consideration because previous research shows that higher levels of psychological difficulties may be experienced prior to or subsequent to dropout among children ([31]) and that youth perceive youth sports as overly competitive and elitist ([27]), meaning some youth prefer to leave sports. Such results indicate that some youth gain health and resilience in sports at the expense of others and that youth leave sports for self-protection reasons ([26]). In the present study, we do not know how the young people’s mental health would have developed if quitting was not an option (as in the case of physical education in school). The need for more studies in this field is obvious. We generally know too little about what is required in terms of mental resources to remain in sports and whether it is the lack of such resources that leads young people to drop out of sports. We need person-centered analyses that can follow young people over time and predict who will remain in sports and benefit from being there and who is more likely to quit because staying could harm their health.

### 4.1. Practical Implications

Our analyses indicate that both traditional organized youth sport contexts and informal, negotiable, and often self-organized movement contexts may affect important well-being variables that support mental health. However, despite reliable analyses, the results should be interpreted with caution. Based on the results, we suggest that further studies on the mental effects of traditional sports participation are needed. However, from a youth development or mental health perspective, it is even more beneficial to study how the facilitation of traditional Olympic youth sports and more informal, negotiable, and perhaps self-organized movement contexts can together contribute to improving developmental environments where children and youth grow up. This seems to be incredibly important, as prior research shows bidirectional relationships. Thus, the design and implementation of youth sport programs should maximize mental health benefits, and the programs should be designed and implemented to attract participants with poor psychosocial health ([32]). Prior research shows that informal and often self-organized movement contexts ([28]) are anchored in different logics with different aims that are negotiable ([24]) and may thus be more profitable for those who do not fit into formal and competitive youth sports. It is reason to follow up on the research from Bean and colleagues ([2]) and others ([11]; [28]; [32]) who claim that to fully succeed in building developmental assets through contexts based on movement activity, programs must be structured for this purpose. This means that if the aim is to support mental health among young people, governments and municipalities need to explore movement contexts and programs that can be developed to enrich the environment with the purpose of generating well-being, quality of life, and continuing motivation among youth. When such programs are developed, Olympic youth sports can be maintained and remain as is: not for all ([26]), yet of significant importance for some.

### 4.2. Strengths and Limitations

The present study is one of the few studies that have examined changes in mental states in light of sport participation using a longitudinal repeated measures approach. Due to the longitudinal design, the time period of two years between the two data waves, the solid sample size, and the ability to compare change between three groups representing three different types of individual actions, there is reason to consider the present study as a reliable and valid study. However, despite the appropriate design, it should be emphasized that the data examined in this study were not collected for the purpose of this study, and because of that, there are nuances that have not been considered. Prior research shows that different sport contexts affect mental health differently and that female and male participants react differently to different contexts ([11]). There is reason to believe that our analyses would have been more exact if a variety of sports and sport contexts were identifiable. Another limitation is that our data did not allow us to control for the reason why some of the youth in the sample dropped out of organized sports. This means that we do not know the exact cause of the reduced well-being among those who dropped out of sports and did not pick up an informal and self-organized activity.

## 5. Conclusions

There is an overall agreement across nations and sectors that mental health issues in youth are detrimental to those who suffer directly as well as to society; that it is important to prevent such health problems in every way possible; and that under normal circumstances, it is indeed possible to strengthen the mental health of young people. This study confirms a negative trajectory in mental health and well-being during adolescence but finds no reason to claim an overall association between youth sport participation and psychological distress symptoms. Our study confirms prior studies that show a rather complex relationship between participation in traditional youth sports and mental health. Dropout of sports during adolescence may have a negative effect on well-being, unless the formal and traditional sport contexts are replaced with more informal and thus negotiable movement contexts. The results presented confirm the need for further exploration of how social contexts based on movement activity can be applied for enriching environmental networks around children and youth.

## Figures and Tables

**Table 1 behavsci-15-01665-t001:** Descriptives.

Variable	Full Sample M (SD)	Dropouts Group 1 M (SD)	Convertors Group 2 M (SD)	Stayers Group 3 M (SD)	Correlation (r)					
					1	2	3	4	5	6
1. Distress Symptoms T1	1.51 (0.53)	1.56 (0.61)	1.57 (0.53)	1.42 (0.47)		0.53 *	0.49 *	0.36 *	−0.25 *	−0.17 *
2. Distress Symptoms T2	1.64 (0.67)	1.72 (0.73)	1.79 (0.73)	1.56 (0.62)			0.33 *	0.54 *	−0.13 *	−0.22 *
3. Negative affects T1	2.96 (1.19)	2.79 (1.01)	3.10 (1.16)	2.81 (1.12)				0.36 *	−0.06	−0.11 *
4. Negative affects T2	3.03 (1.33)	3.20 (1.38)	3.19 (1.36)	2.91 (1.29)					−0.14 *	−0.07
5. Positive affects T1	5.11 (1.13)	4.89 (1.22)	4.91 (1.14)	5.28 (1.05)						0.41 *
6. Positive affects T2	5.06 (1.22)	4.60 (1.26)	4.83 (1.27)	5.23 (1.17)						

Note: * For each correlation, a Credibility Interval was estimated. If the 95% CI does not include zero, the null hypothesis was rejected as improbable, and the correlation is considered credible.

## Data Availability

The original contributions presented in this study are included in the article. Further inquiries can be directed to the corresponding author.

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
