# Peer review of "Temporal Associations Between Sport Participation, Dropout from Sports, and Mental Health Indicators: A Two-Year Follow-Up Study"

_behavsci, 2025, doi:10.3390/bs15121665_

Round 1
Reviewer 1 Report
Comments and Suggestions for Authors
Thank you for the opportunity to review this timely and well-written manuscript. The marked decline in mental well-being among young people is a serious concern and clearly warrants more research into scalable, everyday preventive approaches.
The research question is convincing, the methods sound, and the conclusions reasonable. The paper reads well throughout. I have only minor suggestions for improvement:
Introduction
Line 50: The link between burnout and early specialization is intriguing. One more sentence of elaboration would help the reader understand the mechanism or context.
Line 60 ff: The statement “few, if any, have compared…” is too broad. Given this is central to your work, please support it with citations or note explicitly if no prior studies exist. This strengthens your argument.
Methods
Line 80 ff: Are these results from Erdvik et al., 2019? This section should be more densely referenced; otherwise, parts of it read more like results.
Line 106: Please explain why you chose the Derogatis short form over other instruments. Include brief psychometric data and cite validation studies of the Norwegian version if applicable.
Line 122 ff: Many readers may not be familiar with Bayesian methods. Add a short, accessible explanation of why you chose them and what their main advantages are, something that bridges the gap for a general psychological audience.
Results
Line 150 ff: The latent change score analyses look solid. I suggest adding a clear visual, perhaps a simple plot or figure, to convey the core finding. It would make the results more accessible and memorable.
Discussion
Line 208 ff: Consider adding citations to support the mechanisms you propose.
Line 248 ff: You may want to note explicitly that this is a secondary analysis using data not originally designed for these questions. A short discussion of how this might limit the conclusions would strengthen transparency.
Overall, this is a strong paper; clear, relevant, and thoughtfully executed. With a few clarifications and additions, it will be even stronger.
Reviewer 2 Report
Comments and Suggestions for Authors
Dear authors, thank you for the opportunity to read and learn from your work. I believe that it provides valuable information on the relationship between participation in and dropout from sports and mental health. I consider it essential that research continues along these lines. Below, you will find a series of reviews that may improve your work. Initially, these will be more general in nature, and later more specific and linked to each section.
- It is important to expand on the information about the benefits of organised sports and mental health mental health in the young.
- It is necessary to define fundamental concepts in the work to make it easier to read.
- It would be important to indicate the code number of the bioethics committee that approved the project.
- Expand the discussion with a greater number of studies.
- Indicate the main practical application of the work.
- Define more clearly the future lines of research that should be implemented.

Round 2
Reviewer 2 Report
Comments and Suggestions for Authors
Thank you very much for your responses.
I believe you have done an adequate job with the requested revisions.
I hope your work will be published soon so that knowledge in this field can continue to grow.
Kind regards.